# Opinion Dynamics Explain Price Formation in Prediction Markets

**DOI:** 10.3390/e25081152

**Published:** 2023-08-01

**Authors:** Valerio Restocchi, Frank McGroarty, Enrico Gerding, Markus Brede

**Affiliations:** 1School of Informatics, The University of Edinburgh, Edinburgh EH8 9AB, UK; 2Southampton Business School, University of Southampton, Southampton SO17 1BJ, UK; f.j.mcgroarty@soton.ac.uk; 3School of Electronics and Computer Science, University of Southampton, Southampton SO17 1BJ, UK; eg@ecs.soton.ac.uk (E.G.); markus.brede@soton.ac.uk (M.B.)

**Keywords:** opinion dynamics, econophysics, prediction markets, complex networks, agent-based modelling

## Abstract

Prediction markets are heralded as powerful forecasting tools, but models that describe them often fail to capture the full complexity of the underlying mechanisms that drive price dynamics. To address this issue, we propose a model in which agents belong to a social network, have an opinion about the probability of a particular event to occur, and bet on the prediction market accordingly. Agents update their opinions about the event by interacting with their neighbours in the network, following the Deffuant model of opinion dynamics. Our results suggest that a simple market model that takes into account opinion formation dynamics is capable of replicating the empirical properties of historical prediction market time series, including volatility clustering and fat-tailed distribution of returns. Interestingly, the best results are obtained when there is the right level of variance in the opinions of agents. Moreover, this paper provides a new way to indirectly validate opinion dynamics models against real data by using historical data obtained from PredictIt, which is an exchange platform whose data have never been used before to validate models of opinion diffusion.

## 1. Introduction

Futures contracts have long been used in finance to harness the *wisdom of the crowd* and make predictions about the future value of an asset by exploiting people’s aggregate expectations. Prediction markets are recent forms of futures markets, and although they were originally only meant to forecast the outcomes of important political events, nowadays they are used in several different contexts. For instance, alongside public markets that allow betting on political or sporting events, private prediction markets, which are used by companies such as Google, Intel, and General Electric, gather people’s beliefs about business activities, such as sales forecasts or the likelihood of a team meeting certain performance goals [1,2]. Although prediction markets are often heralded as effective mechanisms to make highly accurate predictions [3,4], it has been shown that they are prone to bias [5] and manipulation [6], and that these phenomena can spread to financial markets with dreadful consequences [6,7].

To better understand prediction markets and, consequently, account for these adverse events, thereby limiting their impact outside the prediction market itself, there is the need for models that realistically reproduce the underlying processes that drive price and opinion formation. This can be achieved by building adequately complex frameworks that can be validated against real-world data. That is, models that are simple enough to be understood and controlled, but complex enough to allow the emergence of realistic and complex dynamics generated by simple interactions between agents. Despite the vast amount of existing models of prediction markets [5,8,9,10], there is limited work focussed on modelling prediction market exchange platforms, arguably the most crucial type of prediction market.

To address this gap, in this paper, we propose a model that matches the empirical properties (often referred to as *stylised facts*) of historical price and volume time series of political prediction markets. To achieve this, we consider a social network where agents possess an opinion about the probability of a given event occurring, and either buy or sell contracts on a prediction market exchange based on their opinions. To model the opinion dynamics process, we use the Deffuant model [11]. Opinion dynamics have long been a subject to which numerous physicists have applied statistical physics tools to gain a better understanding of human interactions and the complex phenomena that emerge from them (see, for example [11,12,13,14,15] for seminal opinion dynamics models and [16] for a thorough review of the topic). Research has also been conducted using opinion dynamics models with binary opinions to model agents in the stock market [17,18,19,20]. Among the many models proposed to describe opinion diffusion in social networks, we follow the Deffuant model, which has three features that make it especially suitable to represent the underlying opinion propagation process that determines prices in prediction markets. First, the Deffuant model considers continuous opinions bounded by arbitrary values. This makes it perfect to describe the diffusion of opinions about the probability of an event to occur, as the opinion can be bounded between 0 and 1 and take any value in this range. Second, since it only has two free parameters, the Deffuant model has the merit of being extremely simple, which allows us to gain deeper insights into what drives price properties in prediction markets. This also guarantees a good degree of realism without having to make assumptions on other parameters in the model, which have to otherwise be fine-tuned, a common (and often necessary) practice for models of financial markets [21,22]. Third, similar to other bounded confidence models of opinion diffusion, in the Deffuant model, only people with similar opinions update their beliefs after interacting, which allows for the possibility of not reaching a consensus at equilibrium. This property enables us to analyse the similarity of our results to historical data, depending on the number of opinion clusters that coexist at equilibrium.

This paper makes three important contributions. First, we introduce a model of prediction markets that uses opinion dynamics in social networks as its underlying mechanism. This model has the merit of being particularly simple since it possesses only two free parameters, which are highly robust at the same time, and it is capable of generating price time series that match the stylised facts of historical data for any combination of the two free parameters. Second, we show that our model reproduces historical data best when opinions are heterogeneous but revolve around a single opinion cluster. This suggests that participants in prediction markets tend to have a similar, yet not identical view on the outcome, especially in markets that have a limited duration. Third, our results support the presence of Deffuant-like opinion dynamics. This contribution helps tackle an important challenge that Sobkowitz posed by arguing that opinion dynamics models are often disconnected from the real world and lack empirical validation [23]. Since his paper, the popularity explosion that social media has experienced has certainly offered a means of compelling validation [24,25]. However, the availability and popularity of such vast datasets resulted in most opinion dynamics models being validated only on social media data, potentially introducing a bias. In this paper, we use market data to provide evidence of continuous opinion diffusion models; specifically, the Deffuant model is compatible with the exchange of opinions in prediction markets. To achieve this, we use data from PredictIt, a political prediction market exchange platform, and show that the Deffuant model provides an excellent representation of the underlying opinion diffusion process in social networks when opinions are scalar.

The remainder of the paper is organised as follows. In Section 2, we define the model of opinion formation and market exchange. In Section 3, we explain the experimental settings that we used to run agent-based simulations of our model and discuss our findings, showing that our model provides a qualitatively good description of historical price and volume time series even in the worst-case scenario. Finally, we conclude with a short discussion and outline future work in Section 4.

## 2. Model

In this section, we describe the price formation model and argue why it is appropriate to represent prediction markets. We start by describing how prediction markets work; we define normalised prices and true probabilities, and then describe the model dynamics in detail.

Prediction markets are time-limited markets in which contracts are traded on the outcomes of a given event, $E_{i}\in \left\{0,1\right\}$, where *i* denotes the *i*-th event, and can take only two values: $E_{i}=1$ if *i* occurs, and 0 otherwise. Markets for which multiple options are available, (e.g., *Who will win the presidential elections?)* can be seen as markets in which there are *N* events, where *N* is the number of candidates running for president. The payoff of a contract on the *i*-th event is 1 if the corresponding event occurs, and 0 otherwise. Let us denote with ${\tilde{\pi }}_{j}\in \left(0,1\right)$ the price of the contract on the event $E_{i}$. In prediction markets, $\sum_{j}{\tilde{\pi }}_{j}=1+\delta$, where δ is the turnaround (e.g., spread, bookmaker fees, etc.); thus, for our analysis, we consider normalised prices $\pi_{i}=\frac{{\tilde{\pi }}_{i}}{\sum_{j}{\tilde{\pi }}_{j}}$, which provide a better representation of the corresponding realisation probabilities. Also, if a prediction market is completely efficient, the normalised price of a security reflects exactly the probability of the corresponding outcome to occur, i.e., $\pi_{i}=p_{i}=P\left(E_{i}=1\right)$, where $p_{i}$ is referred to as *true probability*.

To model the opinion diffusion process, we follow the Deffuant model [11]. We start by considering a population of N agents, who belong to an undirected, unweighted social network G. Without loss of generality, we assume that there is only one event with two possible outcomes, E=0 and E=1. Then, agent j possesses an opinion $o_{j}\left(t\right)\in \left[0,1\right]$, which can take any real value, and corresponds to the subjective probability that the agent attaches to the outcome E=1. The opinion update process is iterative. At each time step, agents may discuss the event with their neighbors, and update their opinions. To model this process, in every round, agent *i* is randomly chosen from the network to discuss with agent *j*, which is, in turn, randomly chosen among agent *i*’s neighbors. If their opinions are too different, they refuse to update their beliefs. More precisely, the agent pair (i,j) interacts only if $|o_{i}\left(t\right)-o_{j}\left(t\right)|<\varepsilon$, where ε is the threshold of this process and can take any real value between 0 and 1. If the agent pair interacts, they update their opinions as follows:(1)$$\begin{array}{c} o_{i}\left(t+1\right)=o_{i}\left(t\right)+\mu \left[o_{j}\left(t\right)-o_{i}\left(t\right)\right]\\ o_{j}\left(t+1\right)=o_{j}\left(t\right)+\mu \left[o_{i}\left(t\right)-o_{j}\left(t\right)\right] \end{array}$$
where μ is the *convergence parameter*, and $\mu \in [0,\frac{1}{2}]$.

In this model, ε represents the *open-mindedness* of agents who will discuss with—or listen to—other agents, only if their opinions are sufficiently close. In this paper, we follow the basic Deffuant model and consider ε constant among all agents, but there exist other versions of this model in which agents have heterogeneous open-mindedness [26,27]. In general, ε affects the number of clusters at equilibrium (i.e., the number of opinions that coexist), while μ drives the convergence time [11].

To reflect the temporal volume dynamics observed by Restocchi et al. [28], each agent has a probability $p={\left(T-\tau \right)}^{-\gamma }$ to participate in the market, where T is the duration of the market (in days), τ represents the number of days elapsed from the beginning of the market (with 0≤τ<T), and γ=2.44 is a scaling parameter estimated on the same dataset [28]. If agent *i* is chosen to participate in the market, they can either buy or short-sell a contract, whose payoff is 1 if E=1 and 0 otherwise. For the sake of simplicity, we assume that there exists only one event with two possible outcomes. Since agent *i* believes that $P\left(E=1\right)=o_{i}$, they will buy a contract only if the current price $\pi <o_{i}$, and sell (or short-sell) it if $\pi >o_{i}$. They will neither buy nor sell if $\pi =o_{i}$. Following influential agent-based models of financial markets, we assume that price is purely driven by excess demand. Agents can only buy/sell at the current trading price and contracts are created whenever necessary. The demand of agent *i*, $D_{i}$, is proportional to the distance between their opinion and the price, and is described by
(2)$$D_{i}\left(t\right)=o_{i}\left(t\right)-\pi \left(t\right)$$

That is, the more mispriced the agent believes the contract is, the more they will trade. The excess demand (ED) is simply the sum of each agent’s demand, multiplied by a noise term ν∼N(0,σ), as follows:(3)$$ED\left(t\right)=\left|\nu \right|\sum_{i}D_{i}\left(t\right)$$
where, following [29], $\sigma^{2}=0.05$. The reason we model the process with a multiplicative (cf. additive) noise term is that, to ensure that the price does not go beyond the boundaries too often, we need a quenching term. To ensure that this decision does not affect the model’s robustness, we run extensive simulations where the equation for the excess demand is $ED\left(t\right)=\beta \sum_{i}D_{i}\left(t\right)+\nu$. We find that, for 0<β<0.25, the two models return qualitatively similar results. However, adding such a quenching term would require us to run additional calibration and optimisation, eventually risking overfitting our model. By adding a multiplicative noise, we avoid this issue. Importantly, below, we will show that the empirical properties of the time series obtained by running our simulations heavily depend on μ and ε, suggesting that the noise term has little impact on the emerging properties of the model. At each trading round, the price is updated depending only on ED(t). Since prices are bounded between 0 and 1, they are set to 0 or 1 if they become less than 0 or greater than 1, respectively, following an update. Therefore, π(t+1) takes the following values:(4)$$\pi \left(t+1\right)=\left\{\begin{array}{cc} \min\{1,\pi (t)+ED(t)\} & ED(t)\geq 0\\ \max\{0,\pi (t)+ED(t)\} & ED(t)<0 \end{array}\right.$$
where, without loss of generality, we include the equation for the price update for ED=0 in the first case (i.e., ED≥0).

## 3. Results

In this section, we describe the experimental setting that we used to run agent-based simulations of the model, and the results we obtained from such experiments. The simulations were run with all possible combinations of μ and ε, which were the only two free parameters in our model, within the ranges 0≤μ≤0.5 and 0≤ε≤1, which represent the whole possible space for the Deffuant model, with a precision of 0.02 units. Our results show that this model provides a particularly accurate and robust description of prediction markets, because even under the worst-performing conditions, the synthetic time series produced by our simulations capture (at least to some degree) the emerging properties of prediction markets, such as volatility clustering and the absence of the autocorrelation of returns [30].

We tested the model on three different network topologies: a random network, a scale-free network, and a complete graph. We show the results obtained on a complete graph (i.e., any agent is free to exchange opinions with any other agent), since we find that these replicate historical data more accurately. However, we do not find any significant difference in the qualitative behaviours of the results, suggesting that (i) price formation does not heavily depend on the network structure and that (ii) our model is robust to topological changes. Results on the other two network topologies are shown in Appendix A.

Although it has been demonstrated that, at least under some circumstances, diffusion processes display finite-size effects [14,31], we ran experiments with 1000 agents to reflect the scarce liquidity that prediction markets usually exhibit [28], with a median of approximately 300 trades per day per market. In fact, if any finite-size effect exists, this must be displayed by real prediction markets; therefore, by keeping the number of participants low, we can capture such an effect in our simulations. We will leave a detailed analysis of the relation between the network size and price formation in prediction markets in future work. The choice of using Barabasi–Albert networks for our model is motivated by the fact that social networks exhibit hubs and that network topology does not have an influence on the equilibria of the Deffuant model [32].

To specify the other parameters of the simulations, we follow references [28,30], where the authors provided a comprehensive quantitative analysis of the empirical properties of prediction markets using the same dataset from PredictIt. Specifically, for each combination of μ, ε, we ran M=3385 simulations, which was the number of markets used by Restocchi et al. [28,30] in their analysis, and the duration T of each market was randomly drawn from the empirical distribution of durations observed among these markets. In this way, our simulations will produce time series, which are directly comparable with historical data on prediction markets. We initialised our agents with a uniform opinion distribution with an average of $\frac{1}{2}$. This is for three reasons: First, these are the initial conditions that were first used in the paper that introduced the Deffuant model, and are the most widely used in practice. Second, we believe that a uniform distribution with an average of $\frac{1}{2}$ constitutes the most “uninformative prior”, and as such, it should not introduce a bias in our model. That is, other assumptions, especially those with different average opinions, could change the results. Given the impossibility of calibrating the initial opinion distribution with the existing dataset, we decided to remove what would have been another free parameter of the model. Third, for the sake of simplicity, we assume that the true probability is equal to $p=P\left(E=1\right)=\frac{1}{2}$ and is constant throughout the market. This assumption reflects the fact that, on PredictIt and other prediction market exchange platforms, it is possible to bet on an event and its opposite (i.e., there is the possibility to buy and sell contracts on the event *Will E happen?* but also on the event *Will E not happen?*).

To find the optimal values for the pair μ, ε, we follow [33] and define the following objective function:(5)$$f=|k_{sim}-k_{emp}\left|+\lambda \right|\alpha_{sim}-\alpha_{emp}|$$
where $k_{sim}$ and $k_{emp}$ represent the kurtosis value of the distribution of the returns of all 3385 markets, for simulated and historical data, respectively. For the second term, rather than using the value of ARCH(1), i.e., the first autoregressive term of the time series, as suggested by Gilli and Winker [33], we use the value of the scaling parameter α that describes the power-law decay of the autocorrelation function of absolute returns. Since the two terms in (Equation 5) can significantly differ in magnitude, to ensure that no component in the objective function outweighs the other, Gilli and Winker suggest that the second term, in our case $|\alpha_{emp}-\alpha_{sim}|$, is multiplied by a constant $\lambda =\frac{k_{emp}}{\alpha_{emp}}$ that rescales its magnitude. For our data set, Restocchi et al. [30] found that $\alpha_{emp}=0.54$ and $k_{emp}=39.31$, from which we derive λ=72.78. We choose to use α, instead of the first-order autocorrelation term *a*, because we believe this gives the calibration a better accuracy. Specifically, we also tried to calibrate the model by using two different functions, namely $f_{2}=|k_{sim}-k_{emp}\left|+\lambda \right|\alpha_{sim}-\alpha_{emp}|+\lambda^{a}\left|a_{sim}-a_{emp}\right|$, where $\lambda^{a}=\frac{k_{emp}}{a_{emp}}$, and $f_{3}=|k_{sim}-k_{emp}|+\lambda^{a}\left|a_{sim}-a_{emp}\right|$, and found that $f_{2}$ exhibits a qualitatively similar behavior to *f*, without adding information. Also, we found that by using $f_{3}$, as suggested by Gilli and Winker, reduces the sensitivity of the objective function to μ and ε, and generates two regions of local minima in which values are not significantly different. These results are shown by Figure 1.

The objective function values computed from our simulations for each pair μ, ε are displayed by the heatmap in Figure 2. It is clear to see from this figure that there exists one region, approximately within the interval (μ,ε)∈[0.04,0.16]×[0.26,0.5], where *f* is significantly lower than throughout the rest of the space. This region also includes the global minimum, which is found in μ=0.1 and ε=0.32.

It is also interesting to note that *f* displays regular behaviour when varying μ and ε. To better visualise this, in Figure 3, we cut the objective function space in slices and show the behaviour of *f* separately, depending on μ (ε), for a few ε (μ) around its optimal value. By observing these two figures, it is easy to see two regularities. First, from the right-hand side plot, one can see that, in the range 0.26≲ε≲0.3, there is a sharp transition from high values of *f* to low ones, and that this behaviour does not depend on μ (apart from μ≈0). Second, the left-hand side plot shows that *f* always has a minimum approximately in the range of 0.04≲μ≲0.1, but, depending on ε, this minimum is more or less pronounced, and it almost disappears for ε≲0.26.

These results suggest that both μ and ε have a significant impact on the objective function, i.e., both parameters contribute to shaping the statistical properties of the time series generated by the model. This is expected, since, within short time horizons, they both contribute toward the emergence of consensus, and the speed at which this happens. For instance, for high values of μ and ε, the consensus is reached too soon, and the generated time series become less accurate, as suggested by the high value of *f* in the region μ≳0.25∪ε≳0.5. However, our results suggest that ε has a greater impact on *f* than μ. Specifically, we observe that there is one region, delimited by ε≲0.26, for which the objective function value becomes particularly high. Interestingly, this is the same value of ε below, where consensus on a single opinion is not reached in the Deffuant model, and two or more opinions coexist at equilibrium [11], suggesting a relation between the accuracy of our model and whether one or more opinion clusters coexist in the underlying opinion dynamics model. These results suggest that the quantitative accuracy of reproducing $k_{sim}$ and $\alpha_{sim}$ heavily depends on whether there is a single opinion cluster existing at equilibrium, and on the time it takes for opinions to converge. Further evidence is represented by the results shown in Figure 4, in which we show the dependence of *k* and α (Figure 4b) on μ and ε.

The results shown in Figure 4a suggest that the kurtosis of the distribution of raw returns is affected both by μ and ε, but loses its dependence on μ for low values of ε, approximately for ε<0.26, i.e., in the region where multiple opinions coexist at equilibrium. However, the dependence on μ when ε>0.5 suggests that when only one opinion exists at equilibrium, the kurtosis highly depends on the time it takes to reach a consensus. Not surprisingly, the shorter the time to reach consensus, the higher the kurtosis, as once consensus is reached, only a few agents trade, and those who trade have low absolute values of demand, since their opinion is closer to the mean. Similarly, Figure 4b shows that α seems to depend mostly on ε, but exhibits a far sharper phase transition around ε=0.5 than *k*, significantly decreasing its value for ε>0.5. Also, Figure 4b shows that, similar to *k*, in the ε>0.5 region, α depends mainly on μ, and its value becomes larger the faster opinions converge.

These results are further evidence that the accuracy of our model depends on whether there is a single opinion cluster or more, except for the region approximately delimited by (μ,ε)∈[0.25,0.5]×[0.5,1], where consensus is reached early in the market and little or no trading exists after a certain point. Finally, Figure 4 shows that, for any pair μ,ε, the time series generated by our simulations exhibit excess kurtosis and volatility clustering, which are the two main features of the prediction market time series we want to replicate. This is important because the optimal values of μ and ε can significantly change, depending on the dataset used for calibration, but our results suggest that our model is capable to reproduce, at least qualitatively, the empirical properties of prediction markets regardless of the value of its parameters. However, it is important to note that combinations of parameters far from the optimum still show the emergence of the discussed stylised facts, but these are quantitatively far off from those observed in historical data.

Figure 5 shows two comparisons between historical data and simulation results, obtained with both the best and worst configurations, found for the pairs with μ=0.1,ε=0.32 and μ=0.48,ε=0.82, respectively. The comparisons are based on the autocorrelation of absolute returns and the probability density function of absolute returns, which are commonly used metrics to describe time series in financial markets [30,34]. From this figure, we observe that both the best and worst configurations generate distributions of returns, which are similar to the historical ones, only with slightly heavier tails. Results displayed in Figure 5 suggest that, whereas time series generated by the best configuration perfectly match the decay slope of the absolute return autocorrelation function, those generated by the worst configuration do not match historical data accurately. Indeed, although in this case the decay of the autocorrelation function exhibits a long tail, its values quickly dropped when the lag considered was greater than 10 days. This is because, for the pair of values μ=0.48,ε=0.82, the underlying opinion dynamics process converges to one opinion cluster far earlier than the end of the market for most market durations. This causes price changes to be very small or zero, in contrast to the beginning of the market, when price changes are larger due to the higher heterogeneity of opinions.

## 4. Conclusions

In this paper, we propose a model that is able to capture salient features of price formation in prediction markets. To achieve this, we propose an exchange market model in which participants are part of a social network, and exchange opinions about the realisation of a particular event following the Deffuant model. Depending on their opinions, agents buy or sell contracts on a prediction market exchange. By running agent-based simulations, we show that our model generates price time series whose statistical properties closely mimic those of historical data on prediction markets. Interestingly, our findings show that even in the worst-case scenario, our model reproduces prediction markets qualitatively, generating price time series that display volatility clustering and fat-tailed return distributions. These results suggest that a model of prediction markets in which agents interact with each other by sharing and updating their beliefs about an event is suitable for representing prediction markets. At the same time, using historical data, our findings corroborate the validity of the Deffuant model as a representation of real-world phenomena, such as opinion formation with continuous opinions. Additionally, our results show that prediction markets are best reproduced by our model when opinions are scattered around a single opinion cluster. Conversely, when two or more opinions exist at equilibrium (ϵ<0.26) or all agents have exactly the same opinion (large values of μ, for which consensus is reached early), the model yields quantitatively worse results. This suggests that, most of the time, single participants in prediction markets do not reach a perfect consensus on the outcome, but still move close to it. This supports empirical evidence of mispricing in prediction markets, which tends to decrease over time and is higher when market duration is shorter [35]. Our results suggest that this might be the consequence of market participants not having enough time to reach consensus, so that their opinions are scattered around a cluster.

However, our model has some limitations that should be addressed in future work. Most importantly, our model does not include shocks. The availability of new information during a financial market can have a significant impact on prices, and this is true for prediction markets too. The analysis of the impact of shocks on the underlying opinion dynamics process and their impact on price would be of great interest to understand how prediction markets can be made more stable, therefore improving their predictive power and reducing market manipulation. Finally, some of the model’s assumptions were made to simplify this initial exploration of the impact of opinion dynamics on price formation in prediction markets. Although all our assumptions were consistent with the literature, we believe that a thorough stress test could improve the understanding of price formation in prediction markets and perhaps lead to the creation of ad hoc opinion dynamics models for prediction markets.

## Figures and Tables

**Figure 1 entropy-25-01152-f001:**
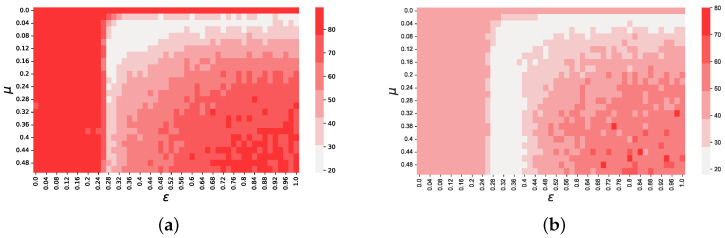
Heatmaps for the objective functions $f_{2}=|k_{sim}-k_{emp}\left|+\lambda \right|\alpha_{sim}-\alpha_{emp}|+\lambda^{a}\left|a_{sim}-a_{emp}\right|$ and $f_{3}=|k_{sim}-k_{emp}|+\lambda^{a}\left|a_{sim}-a_{emp}\right|$, respectively. These figures show that including *a* in the objective function *f* does not add information (**a**), whereas removing α reduces the granularity of *f* (**b**).

**Figure 2 entropy-25-01152-f002:**
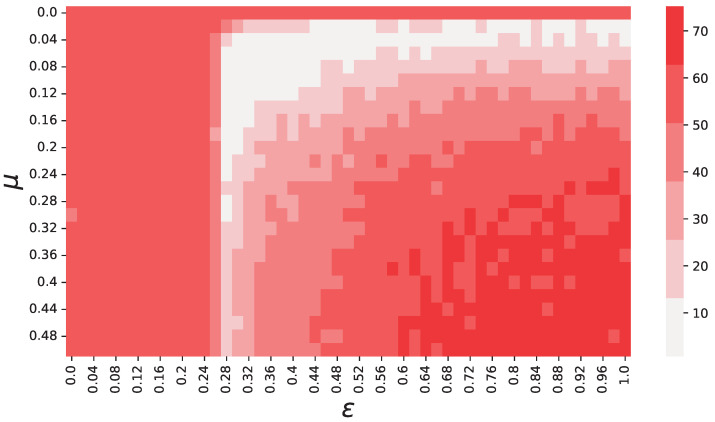
Objective function values (see Equation (Equation 5)) depending on μ and ε. These results show that there is a region, approximately delimited by the area (μ,ε)∈[0.04,0.16]×[0.26,0.5], where the objective function *f* reaches its minima. Each color used in this figure represents an interval of 12.5 for *f*, starting with 0.7<f<13.2. We chose to discretize the colors to smooth our results over noise, and make the regions easily recognizable.

**Figure 3 entropy-25-01152-f003:**
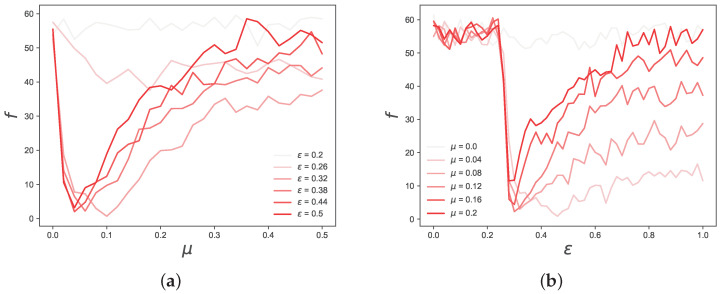
Detailed view of the objective function *f* values, depending on μ (ε), for five values of ε (μ) in the neighborhood of the optimal values, which can be seen in (**a**,**b**), respectively. The objective function shows a strong dependence on both μ and ε, but these figures suggest that ε has a greater impact on *f* than μ. Importantly, from (**b**) we can see that for all values of μ, except for μ=0, there is a phase transition in the value of *f* in the region around ϵ=0.25, which is the threshold after which the Deffuant model starts converging to a single opinion cluster.

**Figure 4 entropy-25-01152-f004:**
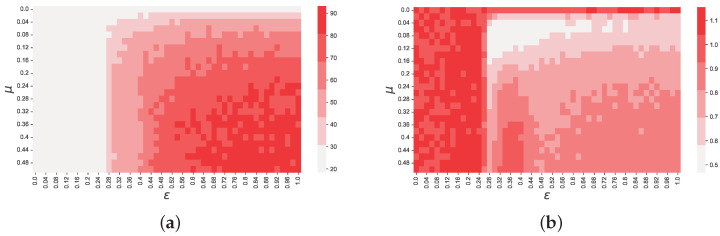
Values of *k* (panel (**a**)) and α (panel (**b**)) depending on μ and ε. These figures show that the kurtosis of the return distribution depends on both μ and ε equally, but the value of α is heavily affected by ε, suggesting a link with the number of coexisting opinions at equilibrium.

**Figure 5 entropy-25-01152-f005:**
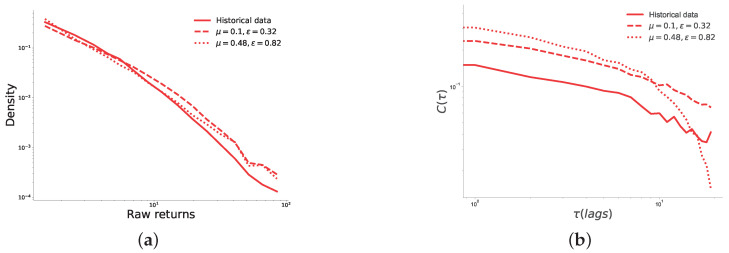
Comparison between historical data and simulation data generated by the best and worst pairs of μ,ε; (**a**) shows the comparison between the probability density distributions of the distributions of raw returns. We note that the distribution of returns spans two decades. This is the maximum range for prediction markets in our data, since the minimum price increment is 0.01 [30], and prices are between 0 and 1. Also, this limits the negative impact of the time series being non-stationary, since large price changes are not allowed by default. (**b**) Shows the decay of the autocorrelation functions. These figures show that both the best and worst fits of our model generate realistic returns, and that the best fit of our model also generates the autocorrelation of returns function that decays with the same exponent as the historical data α=0.54.

## Data Availability

The data were provided by PredictIt and are available for academic research from PredictIt upon request. The code will be made publicly available at www.comses.net (accessed on 25 July 2023) upon publication.

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
