# Peer review of "Opinion Dynamics Explain Price Formation in Prediction Markets"

_entropy, 2023, doi:10.3390/e25081152_

Round 1

Reviewer 1 Report

Current manuscript is well written, with a good organization and language. The message is clear, and the problem is well set and well investigated. 

This manuscript is of a high scientific importance and quality.  Validation of the opinion dynamics is often debatable, yet this piece presents a very interesting perspective. Linking opinion with the probability of events is a very clever trick. I think it might pave the way to much better understanding of both, markets and opinion dynamics. I am of the opinion, that the direction of the research taken, should be further continued. Model seems to capture crucial aspects of reality, in simple yet clever and elegant fashion. Moreover it is supported by data, at least to some degree

There are however some things that should be addressed before publication. I think manuscript lacks  a few examples of simulated trajectories. I would be even better to plot some real world historical trajectories too.

Usage of the network is not sufficiently justified. It would help to compare results for the BA network (used in this study) with alternative ER graph, and a complete graph. It could turn out, that the network component is not necessary at all, and a complete graph topology, like in original Deffuant model, is enough.

Claim:
These results imply that the quantitative accuracy at reproducing

ksim and αsim heavily depends on the number of coexisting opinions at equilibrium,

Claim from lines 232-233 seem to be a stretch, as we observe this only for the region of transition between 1 and 2 clusters, but not between 2 and 3, and further. It is also possible, that In order to detect this dependency one should have a greater resolution for low epsilons and longer times.

I would gladly read this piece again after suggested revisions. Overall, many thanks for the opportunity to review this manuscript. It was a real joy to read it. It needs some polishing, but after that, it may prove a high impact on both fields, mentioned on the beginning of this review.

Reviewer 2 Report

This excellent manuscript attempts to describe price formation issues using the framework of opinion dynamics models.

The authors state that their model (which is based on the bounded confidence model by Deffuant et al.) is able to explain the rich complexity of price dynamics.

I have one minor reservation regarding this manuscript.

I am not an expert in the price formation field. But in the abstract the authors state that "Prediction markets are heralded as powerful forecasting tools, but models to describe them often fail to capture the full complexity of the underlying mechanisms that drive price dynamics"

As such, I expected that the opinion dynamics model will be compared against a model that does not include the social influence effect.

Reviewer 3 Report

I have found this manuscript extremely interesting and commend it on opening an interesting direction on further research of prediction markets. Although I believe there is a lot of room for improvement. Some of suggestions are given below.

The idea of applying opinion dynamics models to model financial markets is not new. Noisy voter model (Kirman model), majority vote model, Galam model were applied before [ https://doi.org/10.1007/s10614-005-6415-1 , https://doi.org/10.1016/j.physa.2011.08.061 , https://doi.org/10.1016/j.physa.2018.10.007, https://doi.org/10.1016/j.chaos.2016.03.011 ]. Interestingly they are all binary state models, and not continuous opinion models.

Furthermore the authors may be interested in an old (but good) overview of agent-based models of financial markets [ https://doi.org/10.3254/978-1-61499-071-0-235 ]. Since then the literature has evolved by much, but there were no critical reviews of the field that I am aware of. Some comparison/overview would be due as one could argue the financial ABMs also model of opinion, despite (usually) not making explicit connections to the opinion dynamics literature.

Lines 95-104 (page 3): Authors should more clearly describe how prediction markets work. Based on some sentences in the paragraph one could assume that an event has only two possible outcomes (e.g., first sentence). While from the others it appears that multiple outcomes could be possible (multiple contract prices). Thinking of the real world examples, would an event be "Will England win FIFA World Cup?" (price for yes, price for no) or "Who will win FIFA World Cup?" (multiple prices for each participating country). Similarly the authors should clearly describe how this different notion of price is represented in the model (because it appears to be just one price; e.g., only for "England will win" event).

The paper lacks explanation on how the trading actually works. Whether the orders are executed via limit order book, or contracts are always bought (sold) by a market maker. Is the number of contracts finite? Or the exchange can create/destroy contracts on the fly?

Lines 125-126 (page 3): With given parameters, p is always larger than 1. Likely the sign of exponent \gamma is to blame. Even if the sign is changed, p would be 1 at \tau=0, and thus all agents would join the market immediately.

What is the unit of time in your model? What is the correspondence to the real time?

What does social network represent in your approach? Typically, in the agent-based models of financial markets no underlying social network is assumed, as the traders do not (usually) directly discuss their opinions with other traders. Traders usually learn by observing order flow.

In the introduction the authors claim that for any value of the two free parameters (\mu and \epsilon) the model generates time series that match the stylized facts of historical data. But judging from the figures it is hardly the truth. Some parameter sets match the empirical data much better than others, while others miss by a lot.

Looking at Figure 3 it seems that the step in parameter space is large than 0.01 (as the authors report). Although in Fig. 1 we see the step size is 0.01. Why the curves in Fig. 3 have less points than expected? E.g. 21 point between \epsilon = 0.0 and \epsilon = 0.2 in Fig 3 (b).

Caption of Fig. 4 has some missing closing parentheses. Also I do not think it is necessary to refer to the same figure within its caption. Simply "(a)" and "(b)" should be enough.

Figure 5 suggests to me that amount of empirical data is severely limited. Raw return distribution is pretty narrow, it spans just two decades. Furthermore I would be doubtful whether plotting the distribution is meaningful as the model is clearly non-stationary. Typically, when analyzing financial market data, or models of financial markets, it is implied that the process (the model) is at least weakly stationary.

Figure 5 (b) suggests that \epsilon is an important parameter, which drastically changes the autocorrelation function of the model. In this context the claims made by the authors, that model replicates the empirical data for all parameter values, seem extremely doubtful. Also I would suggest using log-linear or double logarithmic axes for this plot, as it is not clear whether the autocorrelation curves do not change qualitatively.

Throughout the text the authors haven't specified the initial condition used in their model. As far as I am aware, the outcome of bounded confidence models does depend on the initial condition (initial opinions held by the agents).

Round 2

Reviewer 1 Report

My concerns were addressed carefully and satisfactory. Hereby I recommend following piece for publication.

Reviewer 2 Report

I have no more questions. The authors have addressed all of my comments.

Reviewer 3 Report

The authors have addressed all the issues I have raised. I see no further issues, and thus I recommend accepting this manuscript for publication in its present form.